# The Characteristics and Motivations of Taiwanese People toward Advance Care Planning in Outpatient Clinics at a Community Hospital

**DOI:** 10.3390/ijerph18062821

**Published:** 2021-03-10

**Authors:** Chih-Chieh Yen, Cheng-Pei Lin, Yu-Ting Su, Chiu-Hua Tsu, Li-Mei Chang, Zih-Jie Sun, Bing-Sheng Lin, Jin-Shang Wu

**Affiliations:** 1Division of Hematology/Oncology, Department of Internal Medicine, Douliou Branch, National Cheng Kung University Hospital, Yunlin 640, Taiwan; jack7481@gmail.com; 2Institute of Clinical Medicine, School of Medicine, National Cheng Kung University, Tainan 704, Taiwan; 3Institute of Community Health Care, School of Nursing, National Yang Ming Chiao Tung University, Taipei 112, Taiwan; cp.lin@ym.edu.tw; 4Department of Internal Medicine, National Cheng Kung University Hospital, College of Medicine, National Cheng Kung University, Tainan 704, Taiwan; tracy5270@gmail.com; 5Department of Social Work, Douliou Branch, National Cheng Kung University Hospital, Yunlin 640, Taiwan; joyce@mail.ncku.edu.tw; 6Department of Nursing, Douliou Branch, National Cheng Kung University Hospital, Yunlin 640, Taiwan; pinkn17932003@gmail.com; 7Division of Family Medicine, Department of Internal Medicine, Douliou Branch, National Cheng Kung University Hospital, Yunlin 640, Taiwan; sunzihjie@gmail.com (Z.-J.S.); f22f35f117b2@gmail.com (B.-S.L.); 8Department of Family Medicine, National Cheng Kung University Hospital, College of Medicine, National Cheng Kung University, Tainan 704, Taiwan; 9Department of Family Medicine, College of Medicine, National Cheng Kung University, Tainan 704, Taiwan

**Keywords:** advance care planning, advance decisions, hospice and palliative care, community hospital

## Abstract

Advance care planning (ACP) provides access to complete advance decisions (ADs). Despite the legalization of ACP in Taiwan, it is underutilized in community settings. The objective of this study is to describe the service at a community hospital in Southern Taiwan. We retrospectively analyzed participants who were engaged in ACP consultations from January 2019 to January 2020. The characteristics, motivations, content, and satisfaction of participants are reported. Factors associated with refusing life-sustaining treatments (LST) or artificial nutrition/hydration (ANH) were analyzed using multivariate logistic regression. Of the 178 participants, 123 completed the ACP. The majority were female (64.2%), aged 61 on average and more than 80% had never signed a do-not-resuscitate order. In the ADs, most participants declined LST (97.2%) and ANH (96.6%). Family-related issues (48.9%) were the most prevalent motivations. Rural residence (OR 8.6, *p* = 0.005), increased age (OR 7.2, *p* = 0.025), and reluctance to consent to organ donation (OR 5.2, *p* = 0.042) correlated with refusing LST or ANH. Participants provided a positive feedback regarding overall satisfaction (good, 83%) compared to service charge (fair/poor, 53%). The study demonstrated high AD completion when refusing LST or ANH. These findings may facilitate the development of ACP as a community-based service.

## 1. Introduction

Advance care planning (ACP) is a process that supports healthy individuals or patients in deciding their future care plans and wishes for end-of-life (EOL) [1,2,3]. ACP has received growing attention since the 1990s after the enactment of the *Patient Self-Determination Act* in Western countries [4]. Either as pre-determined or on-demand decisions, ACP assures patient autonomy and medical care consistent with individuals’ preferences and values [5,6]. Valid and informative advance decisions (ADs) provide a solution that balances patient autonomy and physician responsibility when clinical judgments contradict patient preferences. In addition, emerging evidence suggests that ACP improves physical, psychological, and social stress in patients and their family members [7,8,9,10]. Unnecessary medical costs and invasive treatments are also sufficiently reduced, with the lowest risk of jeopardizing rights to medical care [11]. In general, ACP is a proactive, timely, and flexible process that can be applied to routine practices under rapidly changing medical conditions.

Hospice and palliative care services are accessible and mostly reimbursed by the National Health Insurance (NHI) in Taiwan. In terms of gradual improvements and awareness of health and palliative care, Taiwan ranks sixth in the world and first in East Asia in the Quality of Death Index as reported by *The Economist* [12]. The formal and authorized ACP process, as provided by medical institutes and specialized trained personnel, was approved in 2019 after the enactment of the *Patient Rights to Autonomy Act* in Taiwan, which enabled Taiwan to become the first place in Asia to legalize the ACP process and AD records [13]. ACP consultation incorporates a thorough explanation of presumed terminal or potentially agonizing medical conditions, decisions on acceptance or withdrawal of treatments, and the witnessed process of signing the ADs as a legal document. The service can be delivered in acute, intensive, or hospice care wards and is also provided in outpatient clinics in Taiwan [14,15]. Busy inpatient settings, such as emergency units and hospital wards, are not ideal environments for ACP consultations. Outpatient primary care services can be a source of ACP for those with little accessibility to health care, such as home-bound frail elderly [16]. Evidence indicates that decisions made when patients are acutely unwell or medically unstable differ from those of a structured consultation. Therefore, outpatient clinics are considered a better alternative for providing ACP to patients [17,18]. However, there is a limited understanding of ACP delivery at outpatient clinics in community hospitals.

The concept of ACP is largely derived from Western cultures, which leads to substantial obstacles in terms of distinct cultural perspectives. Gallagher et al. reported a survey of 2042 healthy participants in the UK and revealed that dying with dignity was the most frequently discussed and significant issue among all EOL wishes in the ADs [19]. A systemic review indicated that in countries where ethnic diversity is high, such as the US, White patients had a higher acceptability for ACP compared to non-White or ethnic minority patients [20]. In contrast, communication related to EOL issues and acceptance of ACP is relatively low in East Asian countries [21]. For example, in Hong Kong, 81% of frail elderly Chinese patients have never heard of ADs, and 90% of medical students consider themselves undertrained and unprepared for ADs [6,22,23]. A cross-sectional nationwide survey in Japan revealed that only 28.7% of the physicians and 27.6% of the nurses surveyed introduced their patients to ACP [24]. Likewise, ACP is hindered by physicians’ perceptions, family dominated decision-making, and a poor understanding of EOL care [25].

Despite emerging studies confirming the feasibility of ACP consultations, participants’ characteristics, motivations, and outcomes in terms of satisfaction have been under-reported [26,27]. In addition, most of the services are provided in tertiary metropolitan medical centers, where resources are easily available and healthcare awareness is greater compared to rural areas [28]. To improve overall health coverage and promote a broader implementation of ACP regardless of regional disparities, participants have been encouraged to provide ACP consultations in smaller-scale community hospitals. However, the results of these studies have not been explored in detail. Therefore, the objective of this study was to describe the characteristics of individuals who underwent ACP consultations in a community hospital in Southern Taiwan.

## 2. Materials and Methods

### 2.1. Study Design

In the present descriptive study, we retrospectively evaluated volunteer participants who had completed ACP consultations at a community hospital in Southern Taiwan. We elucidated the characteristics, motivations, content of the ADs, and associated factors for life-sustaining treatments (LST) and/or artificial nutrition/hydration (ANH). The definition of terms used in this study is presented in Appendix A.

### 2.2. Participants and Sampling

We enrolled volunteer participants for ACP consultations from January 2019 to January 2020 at the National Cheng Kung University Hospital, Douliou Branch, Yunlin, Taiwan. The study setting was a 200-bed community district hospital located in a suburban region of Yunlin County, which has a population of 0.11 million. In the study hospital during the identical time period, 23.2 cases registered a preference for palliative care at EOL and 122.1 outpatient visits were recorded in the collaborative palliative care clinics as a monthly average. Eligible participants had to be over 20 years of age with a clear consciousness, and voluntarily engage in consultations. We excluded participants if they (1) had signed any ADs previously, (2) had participated in an ACP consultation previously, (3) could not be assured of the self-determining mental capacity, (4) had poor performance status (Eastern Cooperative Oncology Group Performance Status ≥2), (5) refused or failed to cooperate with the screening or formal consultations, and (6) were referred to other hospice or ACP services, or failed to attend the screening interviews or appointments.

### 2.3. ACP Consultations and ADs

The participants first contacted specially trained study nurses via telephone calls or websites to elucidate their demands. The study nurses responded in the form of a pre-consultation interview by telephone or online messages within one week. After screening, eligible ACP participants were scheduled for consultation in formal outpatient ACP clinics.

The ACP consultations were held in an independent outpatient clinic at the study hospital, including a team of three alternating physicians specializing in palliative and hospice care (CC Yen, ZJ Sun, and BS Lin), two study nurses, and two clinical social workers. Specialized training for delivering ACP services is mandatory for all professional staff. The Ministry of Health and Welfare accredited both training courses and formal AD documents. According to the new legislation, the participants were charged an out-of-pocket fee of NT$ 1500 to 3500 (US $51.1 to 120) for each session, which lasted at least one hour. A family member or friend of the participant was required to be present during ACP consultation for the purpose of shared decision-making. The AD documents were printed in traditional Chinese and permitted multiple choices or handwritten descriptions. The English-translated version is provided in Appendix A.

The ACP consultation process included: (1) confirming the participant’s eligibility, preferences, and motivations for ACP; (2) detailed explanations of the five presumed medical conditions: terminal illnesses (e.g., terminal cancer), irreversible comatose status (e.g., severe cerebrovascular stroke), sustained vegetative status (e.g., severe intracranial injury), severe debilitating dementia, and other unspecified agonizing illnesses (e.g., late stage amyolateral sclerosis); (3) choosing to refuse or accept LST or ANH under each of the presumed medical conditions or assigning this decision to a healthcare agent (HCA) recorded in the ADs; (4) assigning EOL preferences and wishes in the ADs; and (5) evaluation of the participant’s understanding of and satisfaction with the session.

### 2.4. Characteristics, Motivations, and Satisfaction

We recorded the clinical and demographic characteristics of the participants, including sex, age, underlying illnesses, residential area, religious background, education, and antecedent preferences for hospice care or cadaveric organ donation prior to ACP. Van Wijmen et al. proposed various possible motivations for participating in ACP [29]. Therefore, we evaluated the motivations mainly derived from three categories: personal, family, or medical service-related issues, by multiple repeated choices after the completion of the ACP session. The case report forms for the investigators are provided in Appendix A. Participant-reported satisfaction parameters, including overall satisfaction, understanding the ACP process, respect for patient autonomy, length of the consultation, environment or personnel, service charge, and others (1–2 = Poor; 3 = Fair; and 4–5 = Good) were assessed immediately after the ACP session and six months post-session via interviews with the study nurses. A satisfactory evaluation form is provided in Appendix A. In addition, we assessed the aforementioned clinical and demographic characteristics to elucidate their correlation with refusing LST or ANH choices based on ADs.

### 2.5. Statistical Analysis

We presented the clinical and demographic characteristics as descriptive analyses in the form of frequencies or percentages. Satisfaction scales were compared as matched and paired continuous variables using a Wilcoxon signed-rank test, with a prespecified two-tailed α of 0.05, to fit in the non-parametric assumption. Dichotomous univariate and multivariate logistic regression analyses were conducted for factors related to accepting LST and ANH in the ADs. All covariates were independent and tested for multiple collinearity and were excluded if violations were detected. Variables with *p*-values less than 0.05, or clinically relevant as determined in the univariate analysis, were selected for the multivariate model. Statistical significance was noted if the covariates had a *p*-value less than 0.05. We used R 3.5.1^®^ (R statistics, Vienna, AT, Austria) and IBM SPSS Statistics 22.0 (SPSS Inc., IL, USA) for data management and computing.

## 3. Results

### 3.1. Participant Characteristics

A flow diagram of the study is exhibited in Figure 1. A total of 178 potential participants contacted the study nurses for ACP consultation. Of these, 154 participants completed pre-consultation interviews. Eventually, a total of 126 participants took part in an ACP consultation, however, only 123 had completed the ADs and were eligible for the analysis and follow-ups, which led to an AD completion rate of 97.6%. The monthly average was 10.5 participants, which was nearly one-twelfth of all outpatient visits in the collaborative palliative care clinics and half of those who registered a preference for palliative care at EOL in the study hospital. The median age of the participants was 60.8 years, 64.2% of whom were female. Most of the participants lived in non-urban regions (suburban, 56.8%; country, 26.0%). The majority of them were married or cohabitating, and approximately half had a college degree or above. Among the participants, only 8.9% had a presenting major illness (malignancy, *n* = 7; rheumatological disorders, *n* = 4) and 65.9% did not have any chronic illnesses. Very few participants had signed wills for palliative care, do-not-resuscitate (DNR) order, or cadaveric organ donation (18.7%, 13.0%, and 11.4%, respectively) (Table 1).

### 3.2. The Motivations for ACP Consultations and ADs

In terms of ADs, most of the participants declined LST (97.2%) and ANH (96.6%) (Table 2). However, in Scenario 1 “terminal illnesses,” a greater number of participants accepted LST (5.7%) and ANH (7.3%). In Scenario 3, “sustained vegetative status,” very few accepted LST (1.6%) and ANH (0.8%) compared to the other scenarios. ANH was significantly preferred in Scenario 1 compared with that in Scenario 3 (Fischer’s exact test, *p* = 0.019). Although a trend toward a preference for LST was observed in Scenarios 1 and 3, it did not reach statistical significance (Fischer’s exact test, *p* = 0.172). Under prespecified conditions, where LST or ANH was conditionally accepted, LST as a time-limited therapeutic trial was requested by six participants with an average time of 3.0 months. Similarly, ANH was requested by eight participants with an average time of 2.6 months. Only one participant requested a HCA to handle medical directives under all scenarios except Scenario 3, where he signed a refusal.

The motivations for ACP consultations are presented in Table 3. Family related issues (48.9%) accounted for the most common drivers to use ACP, followed by personal (27.7%) and medical service-related reasons (23.4%). Only 0.5% reported that assigning a legal HCA (0.5%) was the motivation for ACP (Table 3).

### 3.3. Associated Factors for Refusing LST or ANH

Residential location, reluctance to consent to cadaveric organ donation, and age were significant factors in the univariate analysis. Since religious and educational backgrounds were clinically relevant factors, both were accounted for in the multivariate adjustments. The multivariate regression analysis revealed that non-urban residence (OR = 8.64, *p* = 0.005), increased age (OR = 7.19, *p* = 0.015), and reluctance to consent to cadaveric organ donation (OR = 5.15, *p* = 0.042) were significant independent factors correlated with refusing LST or ANH in this study population (Table 4).

### 3.4. Satisfaction Outcome

From the total study population (*n* = 123), 11 were lost to follow-up at six months, and six participants refused further evaluations of their level of satisfaction with the service. Among those who could be evaluated (*n* = 106), the overall satisfaction feedback revealed a generally high result (good, 83%; fair, 15%) immediately after ACP consultation (Table 5). Details on the satisfaction assessments, such as understanding the ACP process, respecting patient autonomy, the length of time required, and the service environment received good ranks. However, a less satisfactory outcome was observed for the service charge (good, 47%; fair, 23%; and poor, 30%). In addition, free text comments from the participants indicated some difficulties in understanding the meaning of ACP (*n* = 2), the need to simplify the process (*n* = 2), the intention of ACP to protect physicians from medical disputes (*n* = 1), and the futility of ACP in the presence of DNR orders (*n* = 1).

In the median follow-up time of 9.9 months, no participant from the study population had executed care plans based on their ADs. The interview six months after the ACP consultation revealed a trend of a decrease in overall satisfaction (good, 78%; fair, 19%); however, it was not significantly different from the initial assessment (paired Wilcoxon signed-rank test, *p* = 0.104).

## 4. Discussion

The present study illustrated the characteristics of participants and their ADs based on a structured ACP consultation in empirical practice. We demonstrated a high completion of ADs (*n* = 123). To the best of our knowledge, this was also one of the novel studies since the enactment of the *Patient Rights to Autonomy Act* in Taiwan to confirm that delivering ACP services in a community-based institute instead of large-scale medical centers is in fact possible. A comparative study in Germany indicated that the completion of ADs in outpatients did not differ from that in either private practice or university clinics, in which only about one-third of the participants created a living will or assigned a HCA [30]. Mansfield et al. reported that Australian outpatients had a low perception of ACP, with an AD completion rate of 20%, where only 35% appointed an enduring guardian (similar to HCA or a healthcare proxy). This study concluded that ACP was suboptimal in outpatients [31]. Hirakawa et al. further revealed that home-dwelling elderly patients in Japan were reluctant to discuss or complete ADs, made inconsistent decisions as their health condition deteriorated, and were prone to leave EOL decisions to others [32]. Indeed, the utilization of ACP does not seem to be compromised by the scale of the medical institution and is considerably inadequate in homecare or outpatients across diverse cultures. Our results provide information about outpatient ACP services in a community hospital with relatively stable participants.

We observed that the most common motivation for ACP was family related issues. Participants did not want to become a burden on their family and developed their ADs based on their experiences with the death of family members. In addition, ACP frequently involved close family members, as 73.2% of the participants were accompanied by a first degree relative or a spouse. Boerner et al. reported that a well-functioning family and spousal support significantly facilitated discussions on ACP by an increased odd of 2.8 times [33]. Another longitudinal study revealed that problematic family relationships were correlated with a low completion of ACP, and marital satisfaction was a predictor of successful EOL discussions and ADs [34]. Genewick et al. indicated that encouraging family members significantly predicted the completion of ADs for those aged above 50 [35]. Our results, consistent with previous studies, indicate that family dynamics play a crucial role in ACP and should be carefully accounted for by service providers [36].

In addition, we observed that service charge influenced motivation and satisfaction related to the choice to engage in ACP. Of the participants, 9.1% declined the service owing to the payment requirement, and 30.1% graded the charge as “poor.” Despite the wide availability of hospice services in Taiwan’s NHI, it is mandatory to charge a fixed price for ACP consultations [37]. Pelland et al. reported a significant increase in the beneficiaries for ACP after the approval of fee-for-service billing codes in the US Medicare program, suggesting that reimbursement facilitated the utilization of the service [38]. However, such reimbursement may lead to financial incentives to overuse ACP services. A study revealed that Americans are skeptical about Medicare payments for ACP and prefer paying the participants, but not the physicians, to complete ADs [39]. Similar results were observed in two other trials, in which patients’ financial incentives, rather than those of the provider, were correlated with increased ACP utilization [40]. It is critical to strike a balance to facilitate but not overuse ACP services from a financial perspective. Furthermore, a secure fund from the government to facilitate ACP development and implementation has been suggested by an international consensus [2,3,41].

The participants in the present study were well-educated, with a generally high level of well-being, and functionally capable, with only 8.9% suffering from a major medical illness, and none presented with a hastened demand for EOL care at the consultation. Healthy individuals encounter several barriers to ACP compared with chronically ill or debilitated patients, such as prognostic uncertainty, lack of convenient services, younger age, and poor self-health consciousness [31,42,43]. A nationwide cross-sectional study in the UK indicated that only 4.8% of unscheduled hospitalized patients had available ADs [44]. Nevertheless, our results indicate that the ACP service, given the public propaganda and legalization efforts in Taiwan, is still achievable by incorporating relatively healthy people in a proactive manner. In the recent coronavirus (COVID-19) pandemic, as unexpected medical interventions and costs have soared, a clear need has been presented to increase the scope of ACP to maintain stewardship of resources and provide optimal patient autonomy [45,46].

In the present study, the participants presented distinctive values regarding LST and ANH. In the posited medical conditions, LST was not a preference and was regarded as an unnecessary invasive measure for sustaining survival of limited quality. However, ANH has been conditionally accepted as a time-limited trial. The attitude toward depriving hydration or nutrition may have been influenced by culture, since from an East Asian perspective, hunger and thirst are unnatural and agonizing while dying [47]. Furthermore, making a decision to actively discontinue nutrition or fluid support of the elderly is largely recognized as “unfilial” in Asia and leads to guilt imposed by other family members [48]. The acceptance of ANH also varied under different medical conditions. ANH was accepted in terminal illnesses rather than in a sustained vegetative status, with the former integrating it as a part of supportive treatment and not offending the desire for hospice care. Alpert et al. demonstrated that antibiotics and artificial hydration were the least withheld treatments in patients who previously chose to allow themselves to decline. In contrast, cardiopulmonary resuscitation, hemodialysis, and mechanical breathing were the most frequently rejected treatments, similar to our findings [49].

We further observed that participants were more likely to accept LST or ANH if they had consented to cadaveric organ donation. Whether the intention was to facilitate successful donations remains uncertain. A study in Germany revealed that living kidney donors had more adaptive personality traits, higher levels of agreeableness, higher motivation, and lower neuroticism scores than the general population [50]. Despite the growing acceptance in Taiwan, cadaveric organ donation still requires independent consent, which is different from conventional ADs [51]. Another study reported a significant correlation between organ donation consents and successful ADs, however, knowledge about brain death and possible conflicts in ADs was scant [52]. A dilemma is anticipated when “anti-treatment” ADs potentially compromise the optimal condition of the organs for donation [53]. Furthermore, our results demonstrated that those aged over 60 years (OR = 7.19) and non-urban indwellers (OR = 8.64) were prone to refusing LST or ANH. 

Hamel et al. proposed that older age is associated with higher rates of ventilator decline, dialysis, nutritional support, and surgery, even after adjusting for medical conditions and preferences [54]. On the contrary, those who lived in metropolitan cities had more access to medical resources, which potentiated the tendency to accept certain invasive life-prolonging treatments. Although underlying illnesses may be perceived as related to LST or ANH, their association was insignificant, which may be explained by the relatively small number of major illnesses and less suffering from illness in the participants of this study. Together, these results highlighted that the ADs were determined and confounded by participant views on healthcare issues and personal values.

The satisfaction outcomes of the study were generally positive. Respecting autonomy, length of the sessions, and understanding of ACP had received the highest ranks, with the exception of service charges. This reflected the reluctance of the participants toward out-of-pocket payments and their inadequate financial incentives. Zwakman et al. reported that ACP is associated with unpleasant feelings, positive responses, and benefits in hindsight [55]. In the negative comments, misconceptions about ACP included the pro-physician defensive medicine to eliminate disputes, the redundancy in contrast with other hospice wills, and the overt complicated process. Furthermore, the participants’ awareness and satisfaction waned over time. Our results indicated that an overall positive comment on the ACP service and reasonable pricing might lead to a better participant-reported satisfaction outcome.

The merit of this study is that it was conducted in a community hospital where the medical resources are substantially limited and is one of the pioneers in the Taiwan region since the enactment of the new legislation. Further, we provide the participants’ views, including motivation, content, and degree of satisfaction, for further comparisons with other models. However, there are some limitations that should be considered. First, it was a retrospective single-center study with a small sample size that could not address interventional differences. Second, the participants were recruited by online information and represented a selected population with good education, high levels of health consciousness, high socio-economic backgrounds, and high motivation for study participation, which leads to selection bias and prevents extrapolation and generalization. Our future work will focus on expanding the participants to patients from outpatient clinics and inpatient units of oncology service and a comparison of participants from another affiliated urban tertiary medical institute. Third, viewpoints from physicians or other medical staff are not examined for this study. A qualitative exploration among all stakeholders is needed to elucidate the underlying factors that contribute to decision-making. Nevertheless, our results still provide deeper insight and essential information on a structured ACP consultation with high AD completion and illustrate the characteristics of the participants and their will.

## 5. Conclusions

This study demonstrated high AD completion based on structured ACP consultation. The participants engaged in ACP largely due to family related issues. They also declined LST or ANH in general, yet ANH was accepted under some conditions. Non-urban residence, increased age, and reluctance to consent to cadaveric organ donation were associated with refusing LST or ANH. The service received a high satisfaction rank with the exception of out-of-pocket service charges. Together, these results may facilitate the development of ACP as a community-based service, which might be useful in the Asia-Pacific region due to comparable cultural characteristics.

## Figures and Tables

**Figure 1 ijerph-18-02821-f001:**
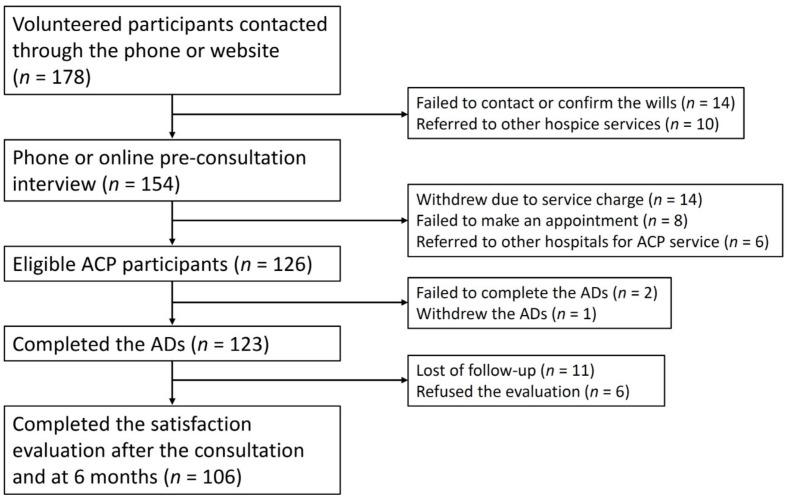
The flow diagram of the study. ACP, advance care planning; ADs, advance decisions.

**Table 1 ijerph-18-02821-t001:** Participant characteristics.

	Participants (*n* = 123)	
Age, years		
Median (IQR)	60.8	(50.0–69.0)
Sex, *n* (%)		
Male	44	(35.8)
Residential location, *n* (%)		
Urban	26	(21.1)
Suburban	65	(52.8)
Country	32	(26.0)
Distance from the study hospital, km		
Median (IQR)	4.8	(2.0–25.8)
Marital status, *n* (%)		
Married or cohabitating	85	(69.1)
Single	19	(15.4)
Unknown	19	(15.4)
Education, *n* (%)		
Primary or below (≤9 grades)	31	(25.2)
Secondary (10–12 grades)	35	(28.5)
College or above	57	(46.3)
Religious background, *n* (%)		
Taoism	44	(35.8)
Buddhism	35	(28.5)
Christian/Catholic	10	(8.1)
Islam	0	
Atheism	34	(27.6)
Underlying illnesses, *n* (%)		
With a major illness ^a^	11	(8.9)
With at least 1 illness	42	(34.1)
Median illnesses (range) per person	2	(0–4)
Illness categories ^b^, *n*	74	
Diabetes mellitus/Metabolic	22	(29.7)
Hypertension/Cardiovascular	18	(24.3)
Osteoarthritis	14	(18.9)
Gastrointestinal	14	(18.9)
Neuro/Psychiatric	5	(6.8)
Companions of the consultation, *n* (%)		
With a 1st-degree relative	90	(73.2)
With an above 1st-degree relative	28	(22.8)
With a friend	5	(4.1)
Antecedent end-of-life preferences ^c^, *n* (%)		
Palliative care at end-of-life	23	(18.7)
Do not resuscitate	16	(13.0)
Cadaveric organ donation	14	(11.4)

^a^ Malignancy (*n* = 7), systemic lupus erythematosus (*n* = 1), Sjögren’s syndrome (*n* = 1), and rheumatoid arthritis (*n* = 2). ^b^ Chronic illnesses according to disease categories, excluding major illnesses. The percentages were provided by the number of illness category/total illnesses. ^c^ According to the registered records approved by the Ministry of Health and Welfare, Taiwan before ACP consultation. IQR, interquartile range.

**Table 2 ijerph-18-02821-t002:** Advance decisions.

	Life-Sustaining Treatments (LST) ^b^		Artificial Nutrition and Hydration (ANH) ^c^	
Scenario 1: Terminal illnesses, *n* (%)				
Refuse	116		114	
Accept ^a^	7	(5.7)	9	(7.3)
Scenario 2: Irreversible comatose status, *n* (%)				
Refuse	121		118	
Accept	2	(1.6)	5	(4.1)
Scenario 3: Sustained vegetative status, *n* (%)				
Refuse	121		122	
Accept	2	(1.6)	1	(0.8)
Scenario 4: Severe debilitating dementia, *n* (%)				
Refuse	119		120	
Accept	4	(3.3)	3	(2.4)
Scenario 5: Other unspecified agonizing illnesses, *n* (%)				
Refuse	121		120	
Accept	2	(1.6)	3	(2.4)

^a^ Including total or conditional acceptance by a prespecified time period or condition. ^b^ LST as a time-limited trial: lasting one month (*n* = 3), three months (*n* = 1), and six months (*n* = 2). ^c^ ANH as a time-limited trial: lasting 0.5 months (*n* = 2), one month (*n* = 2), three months (*n* = 2), and six months (*n* = 2).

**Table 3 ijerph-18-02821-t003:** Motivations for participation.

Motivations ^a^	*n*, (%)	
Family issues	192	(48.9)
Reducing family burden	101	(25.7)
Experience with dying family members	91	(23.2)
Personal issues	109	(27.7)
Personal values of life	88	(22.4)
Pre-existing illnesses	13	(3.3)
Being single or widowed	6	(1.5)
For assigning a legalized HS	2	(0.5)
Medical service issues	92	(23.4)
Trusts the ACP service	49	(12.5)
Avoiding wasting medical resources	30	(7.6)
Attracted by the hospital-led advertisements	13	(3.3)

^a^ A total of 393 motivations were selected from 123 participants as multiple choices. HS, healthcare surrogate.

**Table 4 ijerph-18-02821-t004:** Factors associated with refusing life-sustaining treatment or artificial nutrition/hydration.

Independent Factors			Univariate Regression			Multivariate Regression			
	β	*n*	OR	95% CI	*p*	β	OR	95% CI	*p*
**Residential location**									
**Urban**		**28**	**1**						
**Non-urban**	**1.79**	**95**	**6**	**1.73–20.78**	**0.005 ***	**2.16**	**8.64**	**1.89–39.53**	**0.005 ***
**Age**									
**<60 years**		**59**	**1**						
**≥60 years**	**1.3**	**64**	**3.66**	**0.94–14.25**	**0.061**	**1.97**	**7.19**	**1.28–40.27**	**0.025 ***
**Wills for OGD**									
**Yes**		**15**	**1**						
**No**	**1.51**	**108**	**4.55**	**1.18–17.57**	**0.028 ***	**1.64**	**5.15**	**1.06–30.67**	**0.042 ***
Religious background									
Atheism or others		31	1						
Taoism	−1.03	47	0.34	0.04–3.37	0.369	−1.49	0.23	0.02–2.64	0.236
Buddhism	−1.83	35	0.16	0.02–1.42	0.1	−2.26	0.1	0.01–1.19	0.068
Christian/Catholic	−1.20	10	0.3	0.02–5.29	0.41	−1.77	0.17	0.01–4.92	0.303
Education									
Primary or below		31	1						
Secondary	0.89	35	2.44	0.42–14.38	0.323	1.18	3.27	0.39–27.46	0.276
College or above	0.23	57	1.26	0.33–4.85	0.738	1.26	3.51	0.56–22.20	0.182
Marital status									
Single or unknown		38	1						
Married or cohabitated	0.52	85	1.69	0.50–5.71	0.399				
Companions of the consultation									
1st-degree relatives		90	1						
Others	0.35	33	1.41	0.40–5.05	0.594				
Underlying illnesses									
No		81	1						
Yes	0.49	42	1.63	0.42–6.35	0.485				
Gender									
Male		44	1						
Female	−0.12	79	0.89	0.25–3.13	0.853				
Wills for hospice care									
No		100	1						
Yes	0.15	23	1.17	0.24–5.73	0.849				

OGD: cadaveric organ donation; OR: odds ratio. * denotes a significant level of *p* < 0.05. Bold denotes statistically significant factors.

**Table 5 ijerph-18-02821-t005:** Satisfactory feedback.

	Satisfactory Grading, *n* (%) ^a^			Average Scores
	Good ^b^	Fair	Poor	Mean ± SD ^c^
Overall satisfaction				
After the consultation	88 (83)	16 (15)	2 (2)	4.57 ± 0.82
At 6 months	83 (78)	20 (19)	3 (3)	4.51 ± 0.90
Difference, *p* ^d^		−0.06, 0.104		
Details on satisfaction				
Respecting participant’s autonomy	89 (84)	11 (10)	6 (6)	4.54 ± 0.96
Understanding of ACP	88 (83)	13 (12)	5 (5)	4.53 ± 0.92
Time length or schedule of the session	87 (82)	14 (13)	5 (5)	4.51 ± 0.96
Environment of the clinic	86 (81)	13 (12)	7 (7)	4.46 ± 1.03
Service charge	50 (47)	24 (23)	32 (30)	3.54 ± 1.44

^a^ A total of 106 participants completed the satisfactory evaluation. ^b^ All questions were subjected to the following rankings: good (4 or 5 points), fair (3 points), and poor (1 or 2 points). ^c^ A score range of 1 to 5. ^d^ Compared with paired Wilcoxon sign-rank test, *n* = 106. ACP, advance care planning; SD, standard deviations.

## Data Availability

The data presented in this study are available on request from the corresponding author, Jin-Shang Wu. The data are not publicly available due to possible revelation of private legal documents.

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
