# Peer review of "The Characteristics and Motivations of Taiwanese People toward Advance Care Planning in Outpatient Clinics at a Community Hospital"

_ijerph, 2021, doi:10.3390/ijerph18062821_

Round 1

Reviewer 1 Report

Very interesting and innovative study that demonstrates how the relational approach to medical interventions improves the level of acceptance by patients and therefore the beneficial effects of treatment.
Certainly the case history is very selected both from the social and economic point of view.
Therefore it is suggested to the authors to expand the case history and the type of patients to make the usefulness of the study more democratic.

Reviewer 2 Report

The manuscript entitled “The Characteristics and Motivations of Taiwanese People Toward Advance Care Planning in Outpatient Clinics at a Community Hospital” presents interesting issue, but some areas must be corrected.

General:

The manuscript is shabbily prepared and should be corrected to be formatted according to the instructions for authors – e.g. lining, references section, etc

It seems that none of Authors is a fluent English speaker and some sentences are extremely hard to follow. Moreover, sometimes it is even hard to guess the meaning of sentences – e.g. “Despite the approval since 2019 in Taiwan, it is under-utilized in community settings” – what kind of approval do you mean?). The whole manuscript should be corrected by a native English speaker, preferably by the professional agency.

Abstract:

Authors should provide any results of the statistical analysis

Authors should provide specific conclusions – “The findings provide insight for future ACP development and implementation in community settings” – what kind of insight?

Introduction:

Authors should prepare this section not only to be interesting for Taiwanese readers, but to be interesting for international readers. If Authors prepare their manuscript only for their national readers, they should publish it in some national journal. So, Authors should broaden the information presenting here more international data from various countries, not only the Taiwanese ones.

In this section Authors should briefly present what is already known and what are the “gaps” in the scientific knowledge to formulate the aim of their study properly – based on the literature data.

Materials and Methods:

The applied tools (questionnaires?) should be presented in detail. Authors should present the questions that were asked, the form of the questions, information about previous validation of the questionnaire, etc.

Results:

Authors should verify the representativeness of the studied group

Authors should not reproduce in the text data that are already presented in tables

Tables should be stand-alone ones – be able to be understand without reading the manuscript, so Authors should explain everything needed in footnotes, as well as they should properly formulate the titles of tables to be comprehensive.

Instead of Figure 2, Authors should rather present table, to be easier to follow.

Discussion:

Authors should in their discussion include 3 areas: (1) compare gathered data with the results by other authors, (2) formulate implications of the results of their study and studies by other authors, (3) formulate the future areas which should be studied

Conclusions:

Lines 469-471 – should be removed, as they do not present any conclusions, but rather a self-appraisal

Round 2

Reviewer 2 Report

The manuscript entitled “The Characteristics and Motivations of Taiwanese People toward Advance Care Planning in Outpatient Clinics at a Community Hospital” presents interesting issue, but some areas must be corrected.

General:

The manuscript is shabbily prepared and should be corrected to be formatted according to the instructions for authors – e.g. lining, capital letters, etc

Results:

Authors should verify the representativeness of the studied group
